# Perceived Psychological Restorativeness in Relation to Individual and Environmental Variables: A Study Conducted at Poetto Beach in Sardinia, Italy

Monica Bolognesi [1], Enrico Toffalini [1,*] and Francesca Pazzaglia [1,2]

1   Department of General Psychology, University of Padua, 35131 Padova, Italy
2   Interuniversity Research Centre in Environmental Psychology (CIRPA), 00185 Rome, Italy
*   Correspondence: enrico.toffalini@unipd.it; Tel.: +39-049-827-6256

**Abstract:** This study examines how objective, social, and perceived environmental conditions in a blue space are associated with the perception of psychological restorativeness. We collected data between April 2021 and February 2022 at Poetto Beach in Sardinia, Italy. The participants (N = 255) completed a survey about perceived environmental quality, stress, weather, and restorativeness during their stay at the beach. We used linear models to evaluate the association between psychological restorativeness and social, environmental, and weather parameters. We also analyzed the nature of the association between temperature and restorativeness by viewing this relation as both linear and non-linear and by evaluating the differences in restorativeness between winter, springtime, and summer. The results suggested that the participants viewed the beach as psychologically restorative, especially during the winter season. We also found that the number of people that participants came with was negatively associated with perceived restorativeness. Finally, the results from the correlation analysis revealed that people are less stressed if they go to the beach more frequently.

**Keywords:** blue space; restorativeness; mental health

## 1. Introduction

The rapid trend of urbanization and city living [1] is increasingly correlated with psychologically stressful conditions that derive from the main environmental stressors present in urban contexts, including excessive noise and crowding, traffic, and pollution [2,3]. It is clear that the pressure of modern-day westernized living is taking a toll on human quality of life and well-being [4]. Starting from this awareness, to mitigate these negative environmental effects on humans, actions focused on the changing the quality of life are needed [5]. This important topic has also inspired environmental psychology research over the past 20 years, focusing on the role that the environment has on physiological, psychological, and emotional dimension of the individual. Research has demonstrated that the interactions between people and their environment can also produce positive psychological effects that decrease the impact of urban environmental stressors and have restorative outcomes. Kaplan and Kaplan (1989) [6] first defined environment restorativeness as a process of recovering from mental fatigue or low psychological resources to better meet the demands of everyday life. More recent studies have demonstrated that the restoration process is more present in natural environments than in urban environments [7].

From this basis, scientific literature has increasingly focused on the benefits of contact with restorative environments (mainly natural ones) and their effect on individual and collective well-being, cognitive skills, and emotions e.g., [8–11]. Continuous contact with nature has been shown to offer its users the opportunity to change the mental condition imposed by urban life, which improves mental, cognitive [12], physiological, and emotional restoration [13], particularly in individuals with high levels of psychological

stress [14,15]. Furthermore, poor contact with nature is considered a risk factor for physical and psychological disorders [16–18], supporting the relevance of proximity to and contact with natural environments as an intervention to promote health [8].

Two explanatory theories have revealed the physiological and psychological mechanisms that underline the positive effects of nature in different individual dimensions: the Stress Recovery Theory [19,20] and the Attention Restoration Theory [21]. According to Ulrich's SRT, exposure to natural elements reduces perceived stress and physiological arousal and decreases negative feelings, such as anger or fear [22,23]. Restoration from stress positively affects psychological functioning by improving cognitive skills and performance and by increasing positive-toned affects [20].

Kaplan's ART focuses on the relationship between environment and attentional skills and refers to William James's (1892) work [24] on directed attention [25]. According to ART, nature offers the possibility to restore directed attention and to acquire more self-awareness [26–30]. Cognitive restorations are assumed to be based on four environmental factors: being away, extent, compatibility, and fascination [6]. Being away allows the individual to escape from everyday life and includes the possibility of moving towards other life situations that do not require the use of direct attention, which makes people feel that they are far from their routine and usual environment [31]. Extent refers to a setting that has adequate and interesting content to capture the individual's attention for a long enough time frame and is associated with the spatial and temporal dimensions of the environment being visited. Compatibility indicates a match between a person's inclinations and an environment's characteristics [21,23,32]. It also responds to individuals' needs, improves cognitive functioning, and in an evolutionary perspective, is more present in natural than urban settings [21]. Finally, fascination is the most important environment characteristic in the restorative process. It is a type of involuntary and effortless attention that captures the visitor's interest and has no capacity limitations [31]. It also provides cognitive resource restoration by viewing fascinating objects in the environment [23]. Combining any of these environmental characteristics enhances the potential for a place to provide a more complete restorative experience and ensures a better visiting experience [31].

### 1.1. Blue Space as a Restorative Environment

Blue space includes all outdoor surface waters, such as rivers, seas, and lakes, which are among the most restorative natural environments [33] and can be considered useful resources for increasing physical activity and improving psychological health [34]. Although environmental psychology is researching aquatic environments (blue space) compared with green space [33–36], several studies have shown that visiting blue space is associated with good mental health (e.g., [33,37]) and lower levels of perceived stress [38]. In studies on environmental preference, both natural and artificial scenes containing water were associated with higher preferences and higher perceived restorativeness compared with ones that did not contain water elements (e.g., [33]). This scientific evidence supports the importance of proximity to and contact with blue environments for health promotion. Nevertheless, coastal and marine environments have been identified as suffering more rapid degradation and biodiversity loss than any other ecosystems [8,39] and are considered highly vulnerable ecosystems in relation to local and global climate change and environmental quality [40]. According to the 14th Sustainable Development Goals of 2030 Agenda [5] to allow seas and oceans to contribute to the human well-being, it is important to preserve the physical and biological integrity of these particular and strongly vulnerable biomes. Moreover, beaches and coastal parks have faced a variety of changes in climatic and environmental parameters, including changes in water and air quality, air temperature, and strong gusts of wind onshore and offshore. Air and water temperature have been shown to play a key role in psychological restoration; perceived restorativeness is reportedly higher in coastal environments when ambient temperatures are below the average monthly temperatures [34]. Indeed, it is demonstrated that the increase in air temperature is negatively associated to individuals' cognitive performance, through an important decline in both speed and

accuracy cognitive measure [41]. From a physiological standpoint, temperature and air pollution seem to act together to alter the cardiac autonomic functions [42]. Similarly, Ren et al. (2011) [43] found that temperature may interact synergistically to affect heart rate variability (HRV). These findings demonstrate how climate change and environmental parameters may affect on both psychological and physiological dimensions of humans.

Another important variable is the crowding phenomenon in coastal environments; over 200 million people live on Europe coastlines and in coastal cities, which are among the most visited places in the world.

Research suggests that a massive presence of crowds in tourist places such as beaches may affect the quality of an individual's visit and restorative experience. Highly visited locations have been the focus of environmental psychology studies due to the everyday stressors and difficulties with enjoying the access to nature [34,44]. Furthermore, mass tourism and the lack of control of human activities near the coasts are increasingly threatening marine habitat and their ability to supply important resources for humans [5]. It is also demonstrated that environmental variables such as excessive noise, litter, or garbage can contribute to elicit psychological stress and can influence preference and perceived restorative quality in urban and natural environments [45]. Based on these premises, it is clear how changes in environmental quality and climate can affect perceived restorativeness in blue space. Increasing knowledge about this relationship can raise awareness that human well-being cannot be achieved without the protection of the Earth's ecosystem [5].

### 1.2. Research Purpose

Our research aims to improve the public's understanding of the relationship between environmental variations and individuals' experience in marine biomes.

We investigated how objective and perceived environmental conditions in coastal contexts can affect individuals' perceptions of psychological restorativeness and mental health. Specifically, we designed a comparison with Hipp and Ogunseitan's study (2011) [34] that asks whether perceived and objective environmental conditions are associated with different gradients of perceived restorativeness during the visits. Furthermore, we added variables to gain a better understanding of the person/environment interaction in coastal contexts.

First, we considered the relationship between objective and perceived air temperature and perceived restorativeness as both a linear and a non-linear effect.

Second, we tested our hypothesis that perceived crowding and the number of people in a participant's group at the beach can have negative effects on the gradients of psychological restorativeness.

Third, we investigated whether there were significant associations between psychological stress and objective and perceived environmental variations through bivariate correlations. Finally, we distinguished our analyses of winter, springtime, and summer periods to determine if there were different trends in perceived restorativeness depending on the season. With this background, we expected that the level of perceived restorativeness would be sensitive to gradients in perceived air and water quality. We considered previous literature on people's inability to judge environmental quality and weather patterns correctly [46,47] and deduced that it is possible that the objective and perceived environmental conditions and quality are not associated.

## 2. Method

### 2.1. Environmental Context Selected for this Study

Poetto Beach was the location chosen for this research. It is located in the city of Cagliari (Sardegna, IT) and is one of the main sources of entertainment for residents and tourists alike. Poetto Beach extends for approximately 8 km (almost 5 miles) and was elected as the study site due to it being one of the most densely populated zones in the whole Cagliari area. The large coast size allows people to perform a variety of activities, such as aquatic sports (e.g., surfing, windsurfing, skimboarding, kitesurfing, and swimming) and

other physical activities, such as jogging and outdoor sports. Our research was conducted in the first part of the beach, which is split into two different areas called "Marina Piccola" and "Prima Fermata" (Figure 1). They are among the two most crowded areas of the entire beach. A 2022 report estimated over 6 million tourists registered in Sardinia (the location selected for our study) in the summertime [48]. During the summer months a large number of locals and tourists spend time and relax along the seashore of Poetto beach causing a crowding effect. For this reason, one of the main topics we wanted to investigate was the relationship between crowding and perceived restorativeness.

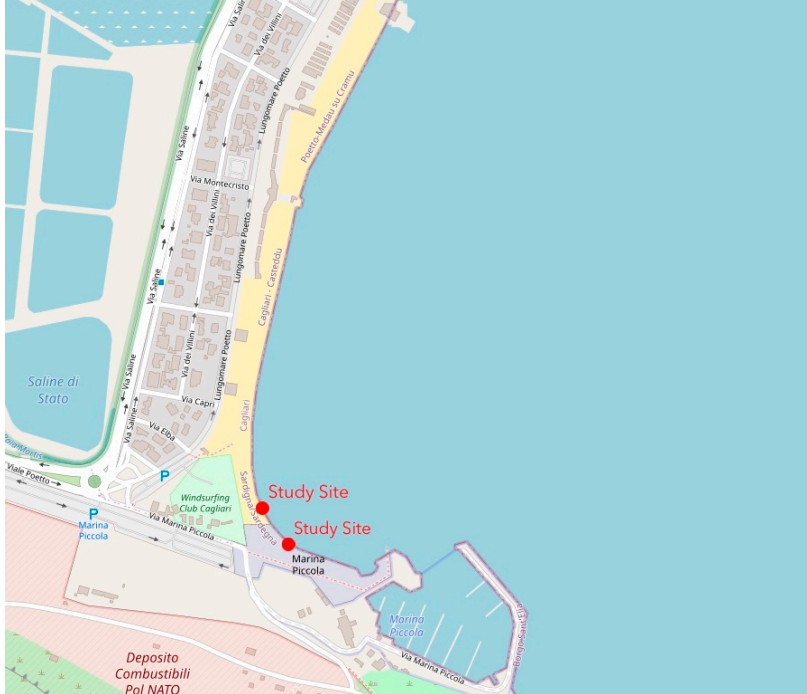

**Figure 1.** Map of the location of study sites [49] (©OpenStreetMap, 2023).

Environmental changes have been extremely visible at the Poetto Beach, especially over the last two decades. In fact, the central-southern coast of Sardinia has a history of water pollution problems caused by sea traffic, particularly in the summertime. Beyond water quality variations, there have also been important air quality changes due to the beach's proximity to urban areas and busy roads. The set of aspects that characterize this blue space led us to believe that Poetto Beach is an interesting location that is suitable for the objective of our study.

### 2.2. Participants

The total number of participants in the current study is 257. Two questionnaires were not returned to the experimenter. Thus, these questionnaires were considered invalid, and the effective sample size included 255 participants. All visitors recruited were 18 years of age or older, with an average age of 39 years. Females were 146 (57%). Basic characteristics of participants and their visit to the beach are summarized in Table 1.

**Table 1.** Descriptive statistics of basic information about participants, their visit to the beach, and social variables.

| Variable | Range | Mean | Standard Deviation | Missing |
|---|---|---|---|---|
| 1. AGE | [18;87] | 39.05 | 15.00 | 0 |
| 2. FEMALE | - | 57% | - | 0 |
| 3. TOTAL FREQUENCY * | [1;3] | 2.84 | 0.87 | 0 |
| 4. DURATION OF STAY * | [1;4] | 2.78 | 0.82 | 0 |
| 5. WEEKLY FREQUENCY * | [1;5] | 2.58 | 1.35 | 0 |
| 6. N OF PEOPLE IN THE GROUP * | [1;3] | 2.38 | 0.67 | 0 |
| 7. RELAX ACTIVITY * | - | 58% | - | 0 |

Note: * See Section 2.3 for clarification on measures marked with an asterisk.

*2.3. Materials*

As mentioned, another important topic of our study is the relationship between environmental and climatic variables and the restorative potential of Poetto Beach.

We used a questionnaire-based survey instrument to test how objective and perceived environmental measurements influence the perceived psychological restorativeness of marine environments (i.e., Poetto Beach). This research focused on the effect of environmental and climatic conditions on psychological restorativeness; therefore, it was necessary to collect both perceived and objective environmental measures.

The Italian survey took its cue from Hipp and Ogunseitan's (2011) research [34], which was conducted in California; however, we made some adaptations to the original questionnaire. For example, we did not measure the tide (both objective and perceived) and we added the current measure of crowding at the beach. The questionnaire was divided into five different parts:

- General Information and Visiting Experience

This part is comprised by questions about individual factors such as age and gender. For each respondent, we also collected information about participant's beach habits and activities. The first question is about how many years has the participant been frequenting the Poetto Beach (total frequency, see Table 1) (approximately, for how long have you been frequenting the Poetto Beach? 1 = one month, 2 = one year, 3 = more than 5 years). The second question is about the duration of the visit to the location (how many hours will you stay at the beach today? 1 = less than 1 h, 2 = 1–2 h, 3 = 2–4 h, 4 = more than 4 h), the third question is about weekly frequency of visitation of participants (how often do you visit this beach per week? 1 = less than once a week, 2 = once a week, 3 = twice a week, 4 = more than twice a week, 5 = everyday), the fourth question is about the number of people in the group that they came with (with how many people have you come to the beach today? 1 = alone, 2 = with another person, 3 = more than another person). Finally, the section ends with a multiple-choice question about which activities the participant expects to do during their visit to the beach that day (1 = relaxing, 2 = socializing, 3 = sunbathing, etc.). In our results, we decided to take into consideration the 3 most performed activities at the beach (see Section 3.1 and Table 1).

- Questions About the Perception of Environmental Conditions at Poetto Beach

In this section, participants responded to questions about their perception of current weather and other environmental conditions (e.g., perceived crowding). The questions were translated and adapted from Hipp and Ogunseitan's (2011) original study [34]. The first and second questions are related to a mensuration of crowding on the beach; participants were asked to estimate how many people they could see from where they were and if they considered the beach to be crowded or not according to their perception. In the other questions, participants reacted to a variety of statements about current environmental and weather conditions, perceived air and water temperature, wind intensity, cloud cover, and air and water quality. Participants' answers were ranked on a Likert-type scale with six different choices, including an "I don't know" option. Responses to questions about air and water temperature ranged from "very cold" to "very hot". All responses with "I don't know" were counted as missing data in the analyses. Water temperature presented

102 (40%) missing responses (presumably because many participants had not probed the water during their stay). For this reason, and because of the strong correlation between perceived air and water temperature among valid responses (r = 0.58), we decided not to consider the perceived water temperature in our subsequent analyses. The perceived cloud cover, which was the only measure recorded in quintiles, ranged from "0%" to "100%". Perceptions of wind were ranked from "no wind" to "strong wind". Air quality and water quality were ranked from "very polluted" to "very clear". Lastly, perceived humidity was ranked from "strong humidity" to "no humidity".

- Perceived Restorativeness Measure

We used the Perceived Restorativeness Scale [50] to measure the perceived restorativeness in beach visitors. PRS is a measure of an individual's perception of psychological restoration in a natural context [51]. This version of PRS is based on Kaplan's ART (1995) [21] and was developed by Hartig, Kaiser, and Bowler (1997) [31]. In the PRS based on Kaplans' ART (1995) [21], the term "extent" was replaced by "coherence" to point out the importance of connectedness with nature and the coherent understanding of the environment. In addition, to better represent the term "extent," items were included to represent the factor legibility, which is a construct related to a visitor's ability to stay oriented and make sense of their surroundings in an unfamiliar environment [52].

In this study, we used the 26 items from the PRS version consisting of 26 statements that measure individuals' perceptions of the five restorative factors previously mentioned. Participants ranked their responses using a 7-point scale, ranging from 0 to 6, to indicate the extent to which the given statement described their experience in a given environment (0 = not at all; 6 = completely). Some negatively worded statements were also included where the value needed to be reversed in coding (e.g., "It is a confusing place").

- Perceived Stress Measure

We included the Perceived Stress Scale [53], which was developed to measure individuals' perception of stress over the last month. The scale presents questions and statements about the degree to which certain situations are uncontrollable and overloaded in daily life. Participants were asked to rank how often they felt a certain way through a 4-point Likert scale. All questions were generic and designed for an audience with at least a junior high school education. In addition, all the possible responses were accessible and easy to understand [53]. For example, "In the last month, how often have you felt nervous and stressed?" The scores were obtained by reverse-scoring the positive items 4, 5, 7, and 8 and summing all the scores. The internal consistency of the results was good (Cronbach's alpha = 0.76).

- Objective Environmental Measures

The objective climatic and environmental quality data were recorded on the day of the survey and represent the data collected closest to the time of the designated visit. The data represented air temperature during each survey period, with monthly averages of the ambient temperature, water temperature, wind speed, cloud cover, and water and air quality. The air temperature was collected from the site Sardegna-Clima (2021) [54]: a non-profit organization that owns 50 weather stations throughout Sardinia. Daily air temperature measurements were compared with the average monthly temperatures collected from ilmeteo [55]. This comparison is important because it detects possible climatic anomalies. Water temperature parameters were collected from the site ilmteteo [55], wind speed and humidity percentage variables were recorded from the site weather [56]. Air quality data were provided by ilmeteo [55], which includes O3, NO2, SO2, CO, PM10, and PM2.5 measures. The site categorically ranks air quality into "excellent", "very good", "good", "discrete", "acceptable", "mediocre", "bad", "very bad", "polluted", and "very polluted".

The quality of bathing water index was recorded from Sardegna ARPA (2021) [57,58]: a regional agency of environmental protection. ARPA annually monitors the quality parameters of both air and water in areas of the Sardinian region. In the monitoring of the years 2021 and 2022, Water quality was always "excellent"; therefore, we did not introduce

it into our analysis. Similarly, air was always in the restricted range between "discrete" and "very good", and the differentiation was largely overlapped by season ("discrete" was registered exclusively in winter days, whereas "very good" was registered in 90% of summer days). Therefore, objective air quality was not used in the analysis and was replaced by season.

### 2.4. Procedure

The questionnaire was used to elicit information about psychological restorativeness in relation to objective and perceived climatic parameters and other environmental gradients. The criteria of inclusion included the following: 18 years of age or older, staying at the beach for the entire duration of the survey, and providing informed consent. The Ethics Committee of the University of Padua has evaluated and approved all materials, questions, and methods (Protocol ID: DFF5921F23E747BA9DCE56AD2C7295B5—4 September 2021).

The survey visit took place at Poetto Beach in the Marina Piccola and Prima Fermata areas. The research required 6 months of recruitment, from April to July in 2021 and from January to February in 2022. Seventy-four participants were recruited in springtime (April through May 2021), 81 in summer (June through July 2021), and the other 100 participants were recreuited in winter (January through February 2022). There were 24 survey visits in total: 11 visits were on a weekday and were on 13 the weekend. This randomized selection of survey dates contributed to providing a variety of climatic conditions, environmental quality parameters, and crowding experiences. A researcher was sent out for two hours for each day of recruitment. Recruitment times were from 12:00 to 19:00.

All respondents were approached directly and asked to voluntarily participate; however, researchers only approached visitors who appeared to be over the age of 18 years old. Prior to presenting the questionnaires, the recruiter asked to every participant if they were 18 years of age or older. Although the questionnaire was self-administered, the recruiter was close to the participants in case of doubt. All questionnaires returned to the experimenter were valid.

Those who decided to join the survey were informed about the research and its aims and each provided informed consent. Participants had the choice of completing the questionnaire as either self- or researcher-administered. Attendees who opted for a researcher-administered questionnaire listened to the surveyor as they read the items without guidance or personal opinions.

### 2.5. Data Analysis

All data analysis was performed using the R programming language [59]. The "lavaan" package [60] was used for performing factor analysis.

#### 2.5.1. Factorial Analysis of Restorativeness

In the first step, we calculated the Cronbach's alpha in each domain of the restorativeness scale (i.e., being away, fascination, coherence, compatibility, and legibility) to ensure that they have good internal consistency. In the second step, we examined whether restorativeness is best represented by a single latent factor. To do so, we performed Confirmatory Factor Analysis (CFA) on the five domains, considered the standardized fit indices (i.e., RMSEA, SRMR, CFI, and NNFI) of the single-factor solution, and measured the reliability by calculating the omega index on the CFA model. We interpreted the good fit indices combined with very good measure reliability (e.g., omega > 0.80) as evidence that a single-factor solution is adequate. We also viewed the standardized loadings. The Akaike Information Index (AIC) was used for every instance of model comparison.

#### 2.5.2. Effect of Temperature on Restorativeness

- Objective air temperature

Unlike other continuous measures, air temperature might have a non-linear effect on restorativeness. Its effect was initially examined under three assumptions. First, we

considered a linear relationship as the simplest and most parsimonious solution and expected an approximately linear decrease in restorativeness with temperature within the range of observed temperatures (i.e., 12 to 32 °C). Second, we compared a non-linear segmented relationship with a to-be-estimated breakpoint (using the "segmented" package of R) [61], which assumes that restorativeness might be stable (or even increase) in the first portion of the range and then drop as temperature increases, but only after a breakpoint. Third, we considered a quadratic effect, where restorativeness might increase in the first portion of the temperature range and then drop with an accelerated speed as temperatures increase. The second and third solutions are obviously less parsimonious than the first one (i.e., they add 2 parameters to the model instead of 1), but they might explain the data better. Both the AIC index and statistical significance were used for every instance of model comparison.

- Perceived air temperature.

Self-reported air temperature was ranked on a 5-point Likert scale, but as explained above, it was collapsed to 4 points. The effect might be linear (decreasing restorativeness with increasingly hot temperatures) or non-linear (presumably with an optimum point around an intermediate level). As perceived temperature was measured on only 4 points, quadratic or segmented regressions as in the previous point would clearly be inappropriate. Rather, non-linearity can emerge treating these 4 points as levels of a categorial/unordered factor. Thus, to detect the best-fitting model, we compared a model that perceived air temperature as quantitative and continuous with a model that treated it as a 4-level unordered factor.

### 2.5.3. Linear Model of Restorativeness

We decided to create two distinct models to predict restorativeness based on objective, social, environmental, and perceived environmental factors. A structural equation modelling (SEM) approach was adopted to provide a comprehensive picture of the predictors of restoration, which simultaneously considers the overall adequacy of the model's standardized fit. A step-AIC procedure was employed to identify the set of predictors in the best-fitting model. With this exploratory procedure, starting from the full model (with all regression paths being estimated), all the alternative models are estimated at zero for one regression path at a time until the AIC cannot be lowered further. "Gender" (treated as a factor) and "age" (treated as a continuous variable) were always entered in the initial models as possible control variables. For the objective, social, and environmental factors, the predictors tested in the linear model included the following: weekly frequency of visits, number of people in the group, season (with two dummy-coded variables: springtime vs. winter and summer vs. winter), objective air temperature, objective cloud cover, and objective wind intensity.

For the perceived environmental factors, the predictor tested included the following: air temperature, air quality, perceived crowding, perceived cloud cover, and perceived wind intensity.

### 3. Results

#### 3.1. Characteristics of Participants and Visit

Basic information of participants is shown in Table 1, which includes the duration of their stay, the weekly frequency of their visits, and the number of people in the group that they came with.

With reference to the data collected, the median weekly frequency of participants was once a week during winter, twice a week during springtime, and more than twice a week during summer. The prevalent duration of a visit was 1–2 h in both winter and summer, and 2–4 h in springtime. Most participants (88%) said that they had been frequenting the Poetto Beach over a period longer than 5 years. The activities that participants performed most of the time while at the beach included "relaxing" and "sunbathing" in summer and "relaxing" and "taking a walk" in winter. In general, the most performed activity was

"relaxing" with 149 answers, follows the activity "taking a walk" with 104 answers and "sunbathing" with 84 answers.

### 3.2. Descriptive Statistics

Descriptive statistics (including number of missing observations) are reported in Table 2. Missing data concerned only perceived factors, with a maximum of 14 missing observations for "perceived humidity". Correlations among all variables of interest are reported in Table 3.

**Table 2.** Descriptive statistics of perceived restorativeness, perceived stress, objective, and perceived environmental variables.

| Variable | Observed Range | Mean | Standard Deviation | Missing |
|---|---|---|---|---|
| 1. BEING AWAY TOTAL (Items 1–5 on PRS) | [0, 30] | 20.12 | 6.59 | 0 |
| 2. FASCINATION TOTAL (Items 6–13 on PRS) | [10, 48] | 34.79 | 8.28 | 0 |
| 3. COHERENCE TOTAL (Items 14–17 on PRS) | [2, 24] | 16.52 | 4.90 | 0 |
| 4. COMPATIBILITY TOTAL (Items 18–22 on PRS) | [2, 30] | 20.89 | 6.58 | 0 |
| 5. LEGIBILITY TOTAL (Items 22–26 on PRS) | [1, 24] | 17.42 | 4.58 | 0 |
| 6. PERCEIVED STRESS TOTAL | [3, 40] | 18.42 | 6.58 | 0 |
| 7. OBJ AIR TEMPERATURE | [12, 32] | 21.53 | 6.78 | 0 |
| 8. OBJ WIND | [5, 30] | 17.13 | 8.37 | 0 |
| 9. OBJ HUMIDITY | [40, 79] | 54.65 | 10.41 | 0 |
| 10. OBJ_CLOUD_COVER | [1, 3] | 1.52 | 0.66 | 0 |
| 11. CROWDING | [1, 800] | 74.79 | 90.09 | 0 |
| 12. PERC CRWODING | [1, 5] | 2.76 | 0.87 | 0 |
| 13. PERC AIR QUALITY | [2, 5] | 4.02 | 0.66 | 12 |
| 14. PERC WIND | [1, 5] | 2.74 | 1.09 | 1 |
| 15. PERC HUMIDITY | [1, 5] | 3.10 | 1.16 | 14 |
| 16. PERC CLOUD COV | [1, 5] | 1.62 | 0.95 | 8 |
| 17. PERC AIR TEMPERAT | [1, 5] | 3.35 | 0.96 | 2 |

Note: For variables 1–6 the observed total scores, calculating adding up the item scores, are reported.

### 3.3. Perceived Stress Correlates

As a preliminary step, we looked at the bivariate correlations between perceived stress and the other variables. All correlations were modest. The strongest ones involved negative associations with weekly frequency of visit at the beach (r = −0.21; $p < 0.001$), and with age (r = −0.24, $p < 0.001$), both of which were of moderate magnitude. All others involved weak associations with |r| < 0.20. Considering restorativeness subfactors, perceived stress presented a weak positive association with being away (r = 0.16, $p = 0.01$), and a weak negative association with coherence (r = −0.14, $p = 0.03$). Other associations emerged with environmental factors, and all were negative: r = −0.13 with objective air temperature, and objective wind speed, and r = −0.16 with perceived humidity (all ps < 0.05). The full set of correlation is shown in Table 3.

### 3.4. Factorial Analysis on Restorativeness

Internal consistency was good in each of the five domains of restorativeness. Being away: alpha = 0.87; fascination: alpha = 0.86; coherence: alpha = 0.77; compatibility: alpha = 0.90; and legibility: alpha = 0.83. A single-factor model with the five domains had very good fit indices, $\chi2(5) = 0.97$, $p = 0.97$, RMSEA = 0.00, SRMR = 0.01, CFI = 1.00, NNFI = 1.02, AIC = 7798.49, and acceptable reliability, omega = 0.78. However, the standardized loadings were not all optimal: for being away, std.B = 0.52; for fascination, 0.74; for coherence, −0.04; for compatibility, 0.94; for legibility, 0.74. Therefore, coherence seems unrelated to the general restorativeness. Fixing the coherence loading to zero, thus considering it as a separate aspect of restorativeness, led to similar fit indices, but improved reliability for the latent factor and a better (i.e., lower) AIC: $\chi2(6) = 1.27$, $p = 0.97$, RMSEA = 0.00, SRMR = 0.02, CFI = 1.00, NNFI = 1.02, AIC = 7796.78, omega = 0.83. Thus, we retained the latter solution as the best one, with a latent factor for restorativeness provided by being away, fascination, compatibility, and legibility. On the contrary, coherence was examined separately. The final factorial solution is shown in Figure 2.

**Table 3.** Correlation matrix of perceived restorativeness, perceived stress, social, objective, and perceived environmental variables. N = 255 (pairwise).

| Variable | 1 | 2 | 3 | 4 | 5 | 6 | 7 | 8 | 9 | 10 | 11 | 12 | 13 | 14 | 15 | 16 | 17 | 18 | 19 | 20 | 21 | 22 |
|---|---|---|---|---|---|---|---|---|---|---|---|---|---|---|---|---|---|---|---|---|---|---|
| 1. BEING AWAY TOT. | - | | | | | | | | | | | | | | | | | | | | | |
| 2. FASCINATION TOT. | 0.42 *** | - | | | | | | | | | | | | | | | | | | | | |
| 3. COHERENCE TOT. | −0.3 | −0.05 | - | | | | | | | | | | | | | | | | | | | |
| 4. COMPATIBILITY TOT. | 0.49 *** | 0.70 *** | −0.04 | - | | | | | | | | | | | | | | | | | | |
| 5. LEGIBILITY TOT. | 0.39 *** | 0.54 *** | −0.03 | 0.70 *** | - | | | | | | | | | | | | | | | | | |
| 6. PERC STRESS TOT. | 0.16 * | −0.05 | −0.14 * | 0.06 | −0.01 | - | | | | | | | | | | | | | | | | |
| 7. AGE | −0.13 * | 0.05 | 0.10 | −0.07 | −0.01 | −0.24 ** | - | | | | | | | | | | | | | | | |
| 8. N PEOPLE GROUP | −0.15 * | −0.16 *** | −0.10 | −0.18 ** | −0.22 *** | 0.05 | −0.13 * | - | | | | | | | | | | | | | | |
| 9. WEEKLY FREQ | −0.11 | 0.16 * | −0.04 | 0.12 * | 0.08 | −0.21 *** | 0.43 *** | −0.13 * | - | | | | | | | | | | | | | |
| 10. TOTAL FREQ | −0.06 | 0.01 | 0.04 | 0.02 | 0.09 | 0.03 | 0.05 | −0.03 | 0.10 | - | | | | | | | | | | | | |
| 11. DURATION STAY | −0.01 | 0.02 | −0.19 ** | −0.04 | −0.10 | −0.12 | 0.17 ** | 0.06 | 0.26 *** | 0.03 | - | | | | | | | | | | | |
| 12. OBJ AIR TEMP | −0.13 * | −0.12 * | −0.18 ** | −0.15 * | −0.06 | −0.13 * | 0.36 *** | −0.04 | 0.40 *** | 0.07 | 0.32 *** | - | | | | | | | | | | |
| 13. OBJ WIND | −0.02 | −0.01 | −0.05 | −0.03 | −0.03 | −0.13 * | 0.08 | 0.09 | 0.07 | −0.12 | 0.15 * | 0.29 *** | - | | | | | | | | | |
| 14. OBJ HUMIDITY | 0.10 | 0.09 | 0.14 * | 0.12 | 0.06 | −0.06 | −0.05 | 0.07 | −0.04 | −0.13 * | 0.10 | −0.30 *** | −0.03 | - | | | | | | | | |
| 15. OBJ CLOUD COV | −0.03 | −0.19 ** | −0.02 | −0.16 * | −0.10 | 0.05 | −0.04 | 0.03 | −0.05 | −0.10 | 0.01 | 0.09 | −0.13 * | 0.02 | - | | | | | | | |
| 16. CROWDING | 0.07 | 0.07 | −0.11 | 0.04 | 001 | 0.03 | −0.12 | 0.08 | −0.17 ** | −0.03 | 0.09 | −0.25 *** | −0.02 | 0.06 | −0.17 ** | - | | | | | | |
| 17. PERC CROWD | 0.02 | 0 | −0.29 ** | −0.05 | −0.05 | 0.06 | −0.06 | 0.09 | 0.01 | 0.03 | 0.28 *** | 0.24 *** | 0.15 * | −0.06 | −0.10 | 0.32 *** | - | | | | | |
| 18. PERC AIR QUALITY | 0.04 | 0.19 ** | 0.11 | 0.11 | 0.18 ** | −0.13 | 0.20 ** | −0.05 | 0.06 | 0.09 | 0.03 | −0.02 | 0.09 | −0.04 | −0.01 | 0.06 | −0.13 * | - | | | | |
| 19. PERC WIND | 0.05 | −0.01 | −0.03 | 0.01 | 0.01 | −0.07 | 0.01 | −0.06 | 0.14 * | 0.01 | 0.14 * | 0.15 * | 0.50 *** | 0.01 | 0.09 | −0.08 | 0.06 | 007 | - | | | |
| 20. PERC HUMIDITY | −0.17 ** | 0.03 | 0.04 | −0.05 | −0.06 | −0.16 * | 0.28 *** | −0.07 | 0.20 ** | 0.05 | 0.13 | 0.05 | −0.01 | −0.19 ** | −0.02 | −0.03 | 0.04 | 0.15 * | 0.09 | - | | |
| 21. PERC CLOUD COV | 0.04 | −0.10 | 0.15 * | −0.06 | −0.03 | −0.11 | 0.07 | −0.09 | 0.04 | −0.02 | −0.02 | 0.06 | 0.01 | 0.27 *** | 0.50 *** | −0.21 *** | −0.27 *** | 0.03 | 0.01 | −0.15 * | - | |
| 22. PERC AIR TEMP | −0.02 | 0.01 | −0.24 *** | −0.01 | −0.04 | −0.05 | 0.28 *** | −0.07 | 0.35 *** | 0.05 | 0.32 *** | 0.69 *** | 0.09 | −0.22 *** | −0.11 | −0.04 | 0.30 *** | 0.01 | −0.05 | 0.13 * | 0.20 ** | - |

Note. * $p < 0.05$, ** $p < 0.01$, *** $p < 0.001$.

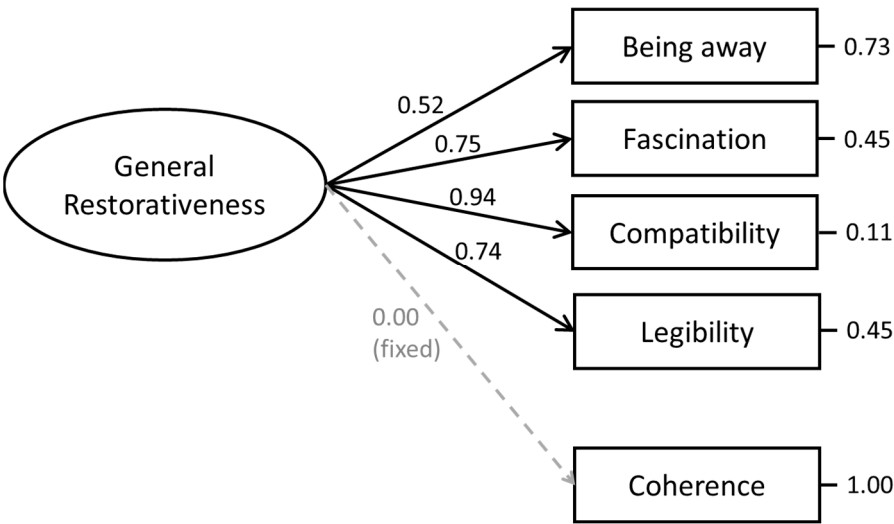

**Figure 2.** Measurement model for restorativeness.

### 3.5. Association between Temperature and Restorativeness

- Objective air temperature

As explained in data analysis, Section 2.5.2, we wanted to establish whether temperature had a linear or non-linear effect on restorativeness. Regarding general restorativeness, a linear model for air temperature presented a significant and negative standardized coefficient std.B = −0.15, $p$ = 0.02, AIC = 1330.23. A segmented model estimated a breakpoint at 24 °C, but its fit was clearly worse than that of the linear model, AIC = 1332.79. Finally, a quadratic model failed to add a significant parameter for the second-degree effect, std.B = 0.04, $p$ = 0.56, and its fit was worse than that of the linear model, AIC = 1331.89. Figure S1 in Supplemental Materials shows a plot of the estimated effects for clarification.

Regarding coherence, very similar results emerged: linear model, AIC = 1530.65; segmented model, AIC = 1531.94; quadratic model, AIC = 1531.73.

In conclusion, for both general restorativeness and coherence, the effect of air temperature was treated as linear.

- Perceived air temperature

Regarding general restorativeness, a model with a linear predictor (AIC = 1325.98) had a better fit than a model with a categorial/unordered predictor (AIC = 1329.95). However, in this case even the former failed to have a statistically significant coefficient, std.B = −0.05, $p$ = 0.83. In fact, a null model with a subjective air temperature coefficient fixed to zero had the comparatively best fit, AIC = 1324.08. Figure S2 in Supplemental Materials shows a plot of the estimated effects for clarification.

Regarding coherence, a different result emerged. The model with a linear predictor (AIC = 1511.09) had a better fit than either a null model (AIC = 2161.71) and a model fit a categorial/unordered predictor (AIC = 1514.30). The linear predictor was negative: standardized std.B = −0.25, $p$ < 0.001.

### 3.6. Structural Equation Modelling

- Objective, social, and environmental factors

The final best-fitting (lowest-AIC; N = 255) model included regressions from the following predictors: For general restorativeness: gender, weekly frequency of visitation, number of people in the group, season (spring and summer vs. winter), and objective cloud cover (final R2 = 0.14). For coherence: age, gender, season (spring and summer vs. winter), and objective air temperature (final R2 = 0.11). AIC of initial model = 7759.03, AIC of final model = 7750.66. Standardized fit indices were very good, $\chi 2(33)$ = 45.23, $p$ = 0.08,

RMSEA = 0.04, SRMR = 0.04, CFI = 0.97, and NNFI = 0.97. Figure 3 below shows the final model with standardized coefficients.

**Objective social and environmental factors**

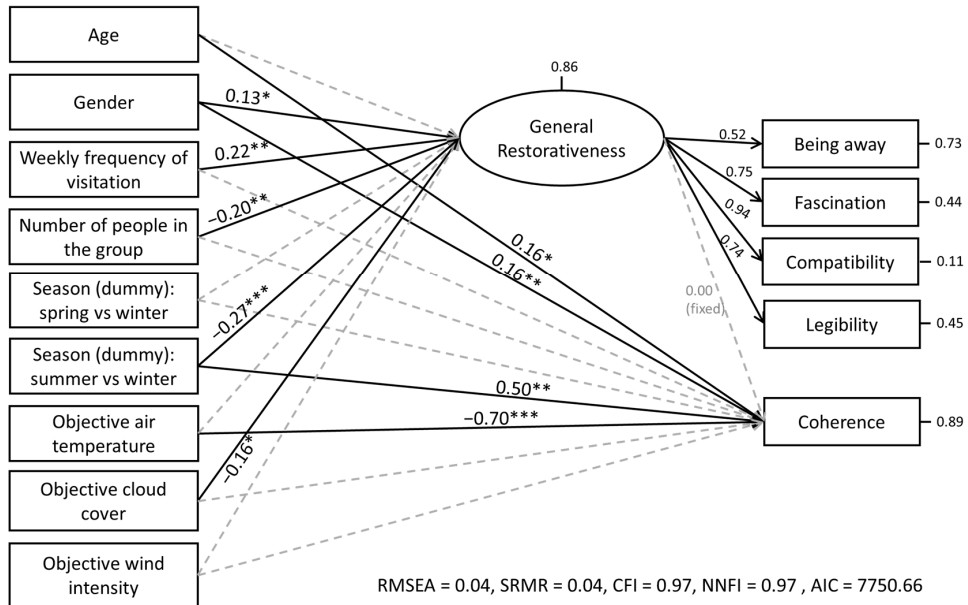

**Figure 3.** Final best-fitting model for objective social and environmental factors. Standardized coefficients are reported. Dashed gray paths are fixed to zero. Note: * $p < 0.05$, ** $p < 0.01$, *** $p < 0.001$.

- Perceived environmental factors

The final best-fitting (lowest-AIC; N = 233) model included regressions from the following predictors: For general restorativeness: gender and perceived air quality (final $R^2 = 0.03$). For coherence: age, gender, perceived air temperature, and perceived crowding (final $R^2 = 0.15$). AIC of initial model = 7090.85, AIC of final model = 7079.50. Standardized fit indices were very good, $\chi^2(35) = 35.32$, $p = 0.45$, RMSEA = 0.01, SRMR = 0.03, CFI = 1.00, and NNFI = 1.00. Figure 4 below shows the final model with standardized coefficients.

**Perceived environmental factors**

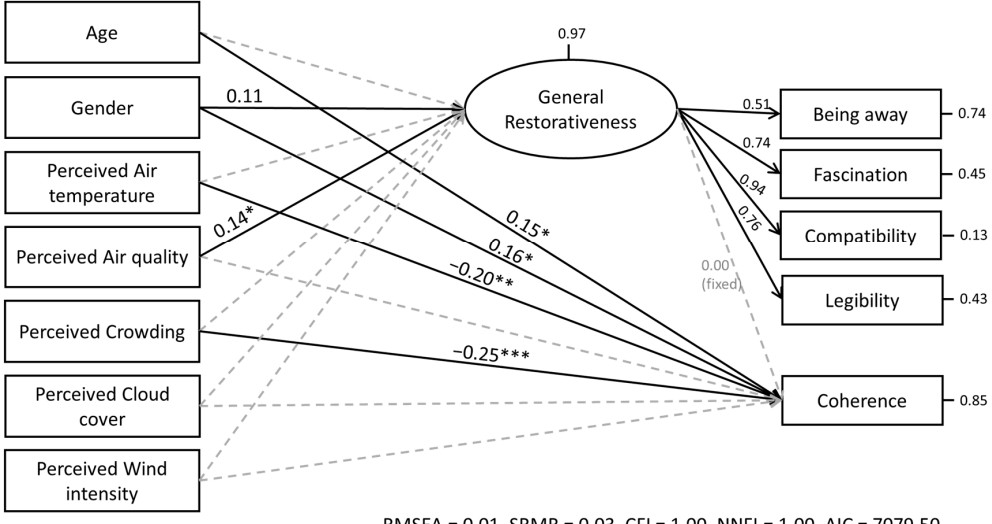

**Figure 4.** Final best-fitting model for perceived environmental factors. Standardized coefficients are reported. Dashed gray paths are fixed to zero. Note: * $p < 0.05$, ** $p < 0.01$, *** $p < 0.001$.

## 4. Discussion

The results presented are interesting, especially considering it is one of the first attempts to establish a link between social and environmental factors and perceived restorativeness. Indeed, we examined the relationship between the environmental, social, and climatic variables, including both the objective and perceived ones and the individuals' experiences in blue space.

Specifically, we aimed to identify the main variables that are associated with enhanced psychological restorativeness. The research was conducted in a Mediterranean context at Poetto Beach in Sardinia, Italy. Previous studies typically focused on the positive effect of blue space in natural and artificial contexts and rarely investigated how perceived and objective gradients of environmental conditions can be related to the experience of psychological restorativeness. Our study was inspired by Hipp and Ogunseitan's (2011) [34] work in California (Orange Coast District), which used equal similar measurements for perceived restorativeness and perceived stress. We adapted the measurements of objective and perceived environmental and climatic conditions for the specific context of blue space in the Mediterranean Sea. For example, we did not include a measure of objective tide (or consequently perceived tide) gradients because there are only minor tide variations for inland seas such as the Mediterranean Sea. In addition, we added a measure of beach crowding, which we anticipated might be relevant in the specific context of a beach close to a major city. Finally, in addition to the weather data, we took season into account due to the very large differences between winter and summer, both in quantitative terms (e.g., minimum and maximum temperature ranges, from min: 6 °C to max: 14 °C in January, and min: 19 °C to max: 30 °C in July) and in qualitative terms (e.g., people engage in different activities in winter vs. summer).

The results from the correlation analysis reveal those who went to the beach more often during the week reported lower levels of perceived stress over the last month. Furthermore, the results from modeling were encouraging by our central thesis about the marked relationship between environmental and climatic conditions on one side and psychological restoration on the other side. Although we treated "Age" and "Gender" as control variables in our study, we observed a moderate association with both perceived restorativeness and coherence. Based on this, being older and female is generally associated with higher levels of perceived restorativeness and coherence (after keeping all other relevant social and environmental factors equal).

Regarding the final model of objective social and environmental factors on general restorativeness, we found that weekly visits are positively associated with perceived restorativeness. In particular, the higher the frequency of visits, the more the environment was rated as psychologically restorative. Furthermore, we found that the perceived restorativeness is inversely associated with the number of people in the group that participants came with. These results can be linked to a higher perceived place attachment to people who visit the site more often. Considering previous literature, the sense of privacy and the need of control and security is important, especially in environments to which one is emotionally attached [62]. This has also been confirmed in more recent studies (e.g., [63]) that demonstrated that higher place attachment is associated to a minor tolerance to crowding. This would explain why perceived restorativeness decreases when the group of people that participant came with is bigger. Another possible explanation, strongly linked to the previous one, can be that other people in the group are perceived as "distraction agents" for their own psychological restoration, but there is no adequate research on this topic. Based on this, in future research it would be important to deepen understand the role of other people in the experience of perceived restorativeness.

Regarding objective climatic conditions, we reported levels of perceived restorativeness are higher in the winter season. Interestingly, it would seem that participants perceived the beach as more regenerative in the colder seasons. We hypothesized that this phenomenon can be explained by the different activities that people perform at the beach during different seasons and by the change in perceived crowding between winter and summer months. However, a causal link between the type of activity and the restorative po-

tential of these factors cannot be made with the current study. According to Self-Regulation Theory, people seek environmental conditions, such as calm and secure places to regulate their affective and psychological states [62]. Visiting the beach in the winter months without crowds can be perceived as a more adequate environment for the self-regulation process.

In addition, only coherence seems to be negatively correlated with perceived crowding. Furthermore, we also observed a weak negative association between perceived restorativeness and objective cloud cover.

Finally, we noticed that air temperature may have a major impact on perceived psychological restorativeness, but the relationship is complex. Past research found that the increase in air temperature causes physiological alterations on individuals (increase in blood pressure and low oxygen saturation in the blood) [64]. It is inferred that the observed physiological effects were mainly responsible for the negative effects on cognitive performance in the same sample [65]. It is also demonstrated that the increases in temperature have a negative effect on the emotional quality of an individual's visit [66]. This last result was partially confirmed in our study. Objective air temperature was discovered to be linearly correlated with overall restorativeness, but this mainly reflected season (e.g., winter observations were associated with higher restorativeness scores than both spring and summer observations). When the season factor was entered in the final model, objective air temperature was no longer relevant.

Regarding perceived environmental variables, we discovered a moderate statistical significance between perceived air quality and psychological restorativeness. As mentioned before, the coherence factor was examined separately. The results of linear modelling revealed that the increase in coherence is associated with warmer seasons. However, the perception of coherence significantly decreases as objective air temperature increases. With respect to linear modelling on perceived environmental factors, air temperature has a negative association with perceived coherence. Coherence is also negatively associated with self-reported perceived crowding. As already mentioned before, past research on place identity found that visitors with higher place attachment and past experience were more sensitive to perceived crowding [63,67]. Considering that the greater part of the participants frequented the Poetto Beach for more than five years, it can be an explanation for the decrease in coherence on days with high gradients of perceived crowding.

Despite the encouraging results, we are also aware that integrating many variables into singular research can have some limitations. For instance, some of the objective weather variables were observed in a too narrow range and did not have an adequate variability to make them meaningful to the analysis (e.g., water quality was always "excellent," air quality was always in the range between "discrete" and "very good" and almost perfectly reflected the season factor). In addition, some variables suffered from collinearity, which made it difficult to disentangle their unique effects from perceived restorativeness. Specifically, air quality was strongly related with air temperature and, in part, with season (see correlations in Table 3), which increases the complexity of understanding the contribution of each individual objective weather variable on restorativeness; for this reason the associated results should be interpreted carefully.

## 5. Conclusions

Despite the limitations mentioned above, the current research is an important step towards developing an understanding of the psychological restorative effects of blue space. It is clear that blue space and its restorative potential receives less attention compared with studies on green space. A possible explanation of this phenomenon can arise from the continuous overlap between blue and green spaces in the study of perceived restorativeness and the presence of complex changes and phenomena in blue space. Due to the morphological and biological complexity of the marine ecosystem, blue space and its restorative potentials cannot be studied in isolation from other environmental factors (e.g., crowding and climatic variations), which can exert a greater influence on behavior change and, consequently, on well-being [8,68].

Due to these premises, we thought it was relevant to investigate the hypothesis that objective, social, and perceived environmental conditions were associated with perceived psychological restorativeness in blue space. To the best of our knowledge, this is the first study to investigate the perceived restorativeness and perceived stress in relation to gradients of environmental quality and climate in a Mediterranean island. The results provide evidence of how environmental factors can affect the quality of an individual's visit, which is especially important because of blue environment's vulnerability to locals and climatic change. More specifically, we found that people who visit Poetto Beach more often perceived lower levels of psychological stress over the previous month. Furthermore, we found that visitors perceived higher levels of psychological restorativeness in colder seasons, which can be seen as an unexpected result. In part, this might reflect the beneficial effect of colder temperatures on restorativeness. However, we believe that this finding also reflects the activities that visitors do at the beach based on the season. In winter visitors go to the beach mainly to relax and take a walk, which are the most restorative activities to do in a natural environment [69]. We also found that psychological restorativeness is positively associated with weekly visit frequency and inversely associated with objective cloud cover and the number of people who came with participants to the beach. Importantly, we also found that perceiving better air quality increases the perceived restorativeness of visitors, which is in part consistent with Hipp and Ogunseitans' findings. For further research, it would be interesting to cross psychological and physical outcomes of the increase in air temperature and air quality in blue space.

As mentioned in the introduction, these current findings reinforce the validity of one of the most urgent topics in the SDGs perspective about marine environment and its vulnerability. It is clear that the link between blue space and human well-being is not one-sided and it is often complex and most of the time human well-being is frequently generated at the cost biome's integrity. For this reason, a change in how humans view, manage, and use marine resources is strongly required [5].

As a conclusion to this work, we recommend to further implement research about blue spaces' vulnerability to climate and environmental changes with methodological improvements and we hope that this scientific evidence can spread awareness about the importance of increasing local and regional sustainable projects, with the aim to promote blue space as a mental health and well-being resource for residents, tourists, and vulnerable people.

**Supplementary Materials:** The following supporting information can be downloaded at: https://www.mdpi.com/article/10.3390/su15032794/s1. Figure S1: Plot of linear (best fit), quadratic, and segmented regression of Restorativeness on Objective Air Temperature. Figure S2: Plot of regression of Restorativeness on Objective Air Temperature with the latter being treated as a linear or category/unordered predictor, as well as a null model including only a constant intercept term (which showed the best fit). Figure S3: Questionnaire. The examples include some items of the following parts: "General Information and Visiting Experience" and "Questions About the Perception of Environmental Conditions at Poetto Beach".

**Author Contributions:** Conceptualization, F.P.; Methodology, E.T.; Formal analysis, M.B. and E.T.; Data curation, M.B.; Writing—original draft, M.B.; Writing—review & editing, E.T. and F.P.; Visualization, E.T.; Supervision, F.P.; Project administration, F.P. All authors have read and agreed to the published version of the manuscript.

**Funding:** This research received no external funding.

**Institutional Review Board Statement:** The study was conducted in accordance with the Declaration of Helsinki, and approved by the Ethics Committee of University of Padua (Protocol ID: DFF5921F23E747BA9DCE56AD2C7295B5—4 September 2021).

**Informed Consent Statement:** Informed consent was obtained from all subjects involved in the study.

**Data Availability Statement:** Research data can be found at the following link: https://osf.io/ra4xd/ (accessed on 13 January 2023).

**Conflicts of Interest:** The authors declare no conflict of interest.

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
