# Peer review of "Perceived Psychological Restorativeness in Relation to Individual and Environmental Variables: A Study Conducted at Poetto Beach in Sardinia, Italy"

_sustainability, doi:10.3390/su15032794_

Round 1
Reviewer 1 Report
Dear Authors
I have completed my review of this manuscript. The study that blue space and psychological recovery is becoming increasingly important. The current research has paid little attention to the relationship between perceived psychological recovery and environmental conditions in marine blue space (beach). The present manuscript provides unique findings on this issue and is therefore of value. However, some modest revisions are needed before further consideration, particularly in the description of the methods and presentation of the results.
Detailed comments are as follows.
Point 1: (The title) Because the manuscript actually discusses psychological recovery, I suggest changing the title to Perceived Psychological Restorativeness in Relation to Individual and Environmental Variables: A Study Conducted at Poetto Beach in Sardinia, Italy.
Point 2: (Line 110-112) Our research aims to improve the public’s understanding of the relationship between environmental variations and individuals’ experience in marine biomes.
This sentence should be in subsection 1.2.
Point 3: (Section 2.1) There should be a map of the location of the study site.
Point 4: (Line 148-149) A 2021 report estimated over 16.5 million tourists registered in Sardinia (the location selected for our study) from June 1 to September 20.
Citations or links should be added.
Point 5: (Line 153-155) Another important topic of our study is the relationship between environmental and climatic variables and the restorative potential of Poetto Beach.
This sentence should be placed at the beginning of the next paragraph.
Point 6: (Section 2.2) These results should be moved to the beginning of Chapter 3. Subsection 2.2 should simply describe how (and by what means) the authors collected information about the participants. Besides, a table/chart should be added to show the basic information about the participants.
Point 7: (Line 194-198) To conclude this first section, we recorded different dimensions of the individuals’ experience at the beach. The first question is about the duration of stay at the beach. The section ends with a multiple-choice question about which activities the participant expects to do during their visit to the beach that day.
The writing style in this section made me not quite understand what the author was trying to express. I suggest that the author rewrite this section or add some details. For example, "The first question is about the duration of stay at the beach." The reader does not have a clear idea of these questions. I suggest that the author show the complete questionnaire in the manuscript/appendix.
Point 8: (Line 210-211) Responses to questions about air and water temperature ranged from “very cold” to “very hot.”
I can understand how the tourists perceived the air temperature, but need to elaborate on the details of perceiving the water temperature. For example, did the author ask these tourists to touch the water body? Or did the author ask the tourists who were involved in sea activities? If neither, then how did the tourists perceive the water temperature?
Point 9: (Line 213-214) Air quality and water quality were ranked from “polluted” to “very healthy.”
Why the opposite term for "polluted" is "very healthy"?
Point 10: (Line 220) PRS is a measure of an individual’s perception of psychological regeneration in a natural context (Pasini et al, 2014).
Regeneration? restoration? recovery? The term should be consistent.
Point 11: (Line 280-281) The research required 6 months of recruitment, from April to July in 2021 and from January to February in 2022.
Due to the difference in survey timing between summer and winter (4 months and 2 months), can the authors supplement the sample size of visitors surveyed separately for summer and winter?
Point 12: (Section 2.4) The following information needs to be added: Are there any invalid questionnaires? What is the validity rate of the questionnaire? What is the rejection rate?
Point 13: (Section 2.5) The authors need to specify the software they used to perform the statistical analysis as well as the structural equation modeling.
Point 14: (Line 334) Why is there a spring season?
Point 15: (Line 362-363) The correlation results are important so I suggest moving Table S1 to the manuscript.
Point 16: (Line 364) S2 is not available in the supplementary material provided by the author, please check the document.
Point 17: (Table 1) The range of items in the Range column regarding ART is confusing, for example the first row BEING AWAY is [0,30]. However, the authors claim that they used a 7-level Likert scale.
Point 18: (Section 3.1) Subsection 3.1 is too brief. The authors mention that "we looked at the bivariate correlations between perceived stress and the other variables", however, only the results of the correlation between perceived stress and visit frequency are reported here. According to S1, perceived stress was also significantly and negatively correlated with environmental conditions (temperature, humidity). In addition, other significant correlation results are worth reporting.
Point 19: (Line 392) Suggest changing the word "for" to "in".
Point 20: (Line 397) Because it is a standardized coefficient, it is recommended to use "SB" instead of "B" for the whole manuscript.
Point 21: (Line 399) what is the total factor represent?
Point 22: (Line 405) The final factorial solution is shown in Figure 2.
Should it be Figure 1?
Point 23: (Figure 1) The title of the figure should be below the figure, please keep it consistent throughout.
Point 24: (Section 3.3) Readers who are not familiar with these methods may not fully understand this section. It is recommended to add pictures/tables of the fit results to show these trends more clearly.
Point 25: (Line 424-425) As regards general restorativeness, a model with a linear predictor (AIC = 1325.98) had a better fit than a model with an unordered factor predictor (AIC = 1329.95).
What is a model with an unordered factor predictor? As in the previous comment, the absence of any explanatory diagrams leads to the fact that readers unfamiliar with these methods may not be able to understand this section.
Point 26: (Line 434) It should be 3.4 subsection?
Point 27: (Line 437) What is the global restorativeness?
Point 28: (Line 442) Should it be Figure 2?
Point 29: (Line 443-451) This part is a duplication.
Point 30: (Figure 3) The title of the figure should be below the figure. Also, according to the information provided by the authors, two separate SEMs should be shown here.
Point 31: (Line 458) The title should be "Social and perceived environmental factors".
Point 32: (Figure 4) Same comments as for Figure 3.
Point 33: (Line 505) Again, what is the global restorativeness?
Author Response
- REVIEWER 1
Dear Authors,
I have completed my review of this manuscript. The study that blue space and psychological recovery is becoming increasingly important. The current research has paid little attention to the relationship between perceived psychological recovery and environmental conditions in marine blue space (beach). The present manuscript provides unique findings on this issue and is therefore of value. However, some modest revisions are needed before further consideration, particularly in the description of the methods and presentation of the results. Detailed comments are as follows.
RESPONSE: We thank the Reviewer for the positive consideration of our paper.
Point 1: (The title) Because the manuscript actually discusses psychological recovery, I suggest changing the title to Perceived Psychological Restorativeness in Relation to Individual and Environmental Variables: A Study Conducted at Poetto Beach in Sardinia, Italy.
RESPONSE: We thank the Reviewer for this suggestion. We have now changed the title as suggested.
Point 2: (Line 110-112) Our research aims to improve the public’s understanding of the relationship between environmental variations and individuals’ experience in marine biomes. This sentence should be in subsection 1.2.
RESPONSE: Following your advice, we have placed the sentence to the subsection 1.2.
Point 3: (Section 2.1) There should be a map of the location of the study site.
RESPONSE: We agree with the Reviewer’s request. We have added a free-copyright, created with OpenStreetMap. We believe that the map will help readers to better understand the location and context of our research.
Point 4: (Line 148-149) A 2021 report estimated over 16.5 million tourists registered in Sardinia (the location selected for our study) from June 1 to September 20. Citations or links should be added.
RESPONSE: We agree that a citation should be added. Unfortunately, the original report could not be retrieved, so we replaced it with the updated 2022 annual report from the Sardinia Region website, and we added a reference to it.
Point 5: (Line 153-155) Another important topic of our study is the relationship between environmental and climatic variables and the restorative potential of Poetto Beach. This sentence should be placed at the beginning of the next paragraph.
RESPONSE: We thank the Reviewer for this suggestion. We decided to place the sentence on paragraph 2.2. Materials, as the explanation of the materials chosen for research.
Point 6: (Section 2.2) These results should be moved to the beginning of Chapter 3. Subsection 2.2 should simply describe how (and by what means) the authors collected information about the participants. Besides, a table/chart should be added to show the basic information about the participants.
RESPONSE: We thank the reviewer for this suggestion. We have now moved part of the subsection 2.2 at the beginning of Chapter 3 (3.1 Characteristics of participants and visit). Regarding subsection 2.2, we have added a table reporting basic information on the participants and their visits to the beach. Further information about how questionnaires were administered is reported in Section 2.4 Procedure.
Point 7: (Line 194-198) To conclude this first section, we recorded different dimensions of the individuals’ experience at the beach. The first question is about the duration of stay at the beach. The section ends with a multiple-choice question about which activities the participant expects to do during their visit to the beach that day.
The writing style in this section made me not quite understand what the author was trying to express. I suggest that the author rewrite this section or add some details. For example, "The first question is about the duration of stay at the beach." The reader does not have a clear idea of these questions. I suggest that the author show the complete questionnaire in the manuscript/appendix.
RESPONSE: We agree that the previous version lacked clarity. As suggested, we have indicated all the questions requested in the first part of the questionnaire (General Information and Visiting Experience in section 2.3), while we did not add the entire set of materials in the manuscript or in the appendix because it is very long.
Point 8: (Line 210-211) Responses to questions about air and water temperature ranged from “very cold” to “very hot.”
I can understand how the tourists perceived the air temperature, but need to elaborate on the details of perceiving the water temperature. For example, did the author ask these tourists to touch the water body? Or did the author ask the tourists who were involved in sea activities? If neither, then how did the tourists perceive the water temperature?
RESPONSE: We thank the reviewer to give us the opportunity to clarify this point. As reported in subsection 2.3 Materials, we specified that participants’ answers were ranked on a Likert-type scale with six different choices, including an “I don’t know” option. Thus, in the question "water temperature" there was the possibility to answer with "I don't know". All responses with "I don't know" were counted as missing data in the analyses. Only responses ranging from “very cold” to “very hot” were considered. Indeed, we decided to add this clarification in the manuscript. Finally, we have explained why perceived water temperature was not considered in our analysis (Section 2.3): as expected, water temperature presented many missing values, and the valid responses were strongly correlated with the perceived air temperature.
Point 9: (Line 213-214) Air quality and water quality were ranked from “polluted” to “very healthy.” Why the opposite term for "polluted" is "very healthy"?
RESPONSE: The original translation from Italian (“aria pulita”) to English was inaccurate. We have amended it as follows: from “very polluted” [“molto inquinata”] to “very clear” [“molto pulita”] (“pulita” literally means “clean”, but when referred to air it can be rendered as “clear”) (Section 2.3 Questions About the Perception of Environmental Conditions at Poetto Beach)
Point 10: (Line 220) PRS is a measure of an individual’s perception of psychological regeneration in a natural context (Pasini et al, 2014). Regeneration? restoration? recovery? The term should be consistent.
RESPONSE: We agree with this suggestion. We changed all the inconsistent words, and we have replaced all the other synonyms with restoration.
Point 11: (Line 280-281) The research required 6 months of recruitment, from April to July in 2021 and from January to February in 2022. Due to the difference in survey timing between summer and winter (4 months and 2 months), can the authors supplement the sample size of visitors surveyed separately for summer and winter?
RESPONSE: We apologize for the lack of clarity in the original version of the manuscript. In fact, organized our data collection to fit observations into 3 seasons of 2 months each (winter, springtime, summer). Data analysis had been conducted according to this categorization from the beginning, although in the original manuscript (and even in the Abstract) we often referred only to “summer” and “winter”. The manuscript has now been revised to make it clear that data were collected in 3 different seasons (section 1.2 and 2.4).
Point 12: (Section 2.4) The following information needs to be added: Are there any invalid questionnaires? What is the validity rate of the questionnaire? What is the rejection rate?
RESPONSE: Thank you for the suggestion. As we have now reported in section 2.4, all questionnaires returned to the experimenter were valid. As previously mentioned in point 6 and in section 2.2 Participants, however, two questionnaires from the original total sample of 257 participants were not returned and were thus considered invalid.
Point 13: (Section 2.5) The authors need to specify the software they used to perform the statistical analysis as well as the structural equation modeling.
RESPONSE: This information has now been added (beginning of section 2.5)
Point 14: (Line 334) Why is there a spring season?
RESPONSE: Thank you for asking. As specified above in point 11, recruitment took place in 3 different seasons: Springtime (April, May) and Summer (June, July) 2021 and Winter (January, February) 2022. Data collection was planned to cover these 3 seasons, and the analysis included season as a category factor with 3 levels since the beginning. This has now been made clearer throughout the manuscript.
Point 15: (Line 362-363) The correlation results are important so I suggest moving Table S1 to the manuscript.
RESPONSE: We thank the reviewer for this important suggestion. We have now moved Table S1 to the manuscript as Table 3.
Point 16: (Line 364) S2 is not available in the supplementary material provided by the author, please check the document.
RESPONSE: We apologize for the mistake. In fact, we mean Table 1, which in the revised manuscript has become Table 2. Thus, we amended the text as follows: “Descriptive statistics (including number of missing observations) are reported in Table 2. Missing data concerned only perceived factors, with a maximum of 14 missing observations for “perceived humidity”.
Point 17: (Table 1) The range of items in the Range column regarding ART is confusing, for example the first row BEING AWAY is [0,30]. However, the authors claim that they used a 7-level Likert scale.
RESPONSE: We confirm that we have used a 7-level (0-6) Likert Scale for PRS. Note that we have reported the total scores of the items for each single sub-factor of the ART, as we have now clarified in a note under Table 2 (Table 1 in the original manuscript). The reported range refer to the minimum and maximum observed values. For instance, the subfactor BEING AWAY is composed by the first 5 items of PRS, so the widest possible range is 0-30, and the observed range in our sample coincide with this range, while for other subfactors it might be different. To make this step clearer, we have added “TOT” for every sub-factors in the currents Tables 2 and 3.
Point 18: (Section 3.1) Subsection 3.1 is too brief. The authors mention that "we looked at the bivariate correlations between perceived stress and the other variables", however, only the results of the correlation between perceived stress and visit frequency are reported here. According to S1, perceived stress was also significantly and negatively correlated with environmental conditions (temperature, humidity). In addition, other significant correlation results are worth reporting.
RESPONSE: This section has now been expanded as requested.
Point 19: (Line 392) Suggest changing the word "for" to "in"
RESPONSE: Thank you for the suggestion. We have amended the text as suggested.
Point 20: (Line 397) Because it is a standardized coefficient, it is recommended to use "SB" instead of "B" for the whole manuscript.
RESPONSE: Thank you for the suggestion. However, for clarity, we preferred to use “std.B” rather than “SB” throughout the manuscript (see section 3.2 and 3.3), but we are open to use the recommended acronym if the Reviewer thinks that it is the better choice.
Point 21: (Line 399) what is the total factor represent?
RESPONSE:
Thank you for this important question. The total factor refers to the second-order “general restorativeness” factor in CFA, that integrates the other subfactors of ART which are Being away, Fascination, Compatibility and Legibility, but not Coherence. To improve the readability, we have decided to relabel “total factor” as “general restorativeness” in a more consistent way throughout the manuscript.
Point 22: (Line 405) The final factorial solution is shown in Figure 2.
Should it be Figure 1?
RESPONSE: Thank you for noticing this. We have now changed “Figure 2” with “Figure 1”.
Point 23: (Figure 1) The title of the figure should be below the figure, please keep it consistent throughout.
RESPONSE: We have now replaced all the titles of the figures as suggested.
Point 24: (Section 3.3) Readers who are not familiar with these methods may not fully understand this section. It is recommended to add pictures/tables of the fit results to show these trends more clearly.
RESPONSE: We have now further clarified the analyses conducted both in section 2.5.2 and in results section 3.3, highlighting that we meant to examine possible non-linear effects of temperature. In addition, a Figure has been included in Supplemental Materials for further clarification (we avoided placing it in the manuscript as it would take away room for more important analyses).
Point 25: (Line 424-425) As regards general restorativeness, a model with a linear predictor (AIC = 1325.98) had a better fit than a model with an unordered factor predictor (AIC = 1329.95). What is a model with an unordered factor predictor? As in the previous comment, the absence of any explanatory diagrams leads to the fact that readers unfamiliar with these methods may not be able to understand this section.
RESPONSE: See the response above. Also in this case, a clarification has been added and a plot of the estimated effects have been shown in Supplemental Materials.
Point 26: (Line 434) It should be 3.4 subsection?
RESPONSE: Yes, we apologize for the mistake. We have now modified Structural equation modelling as 3.4 subsection.
Point 27: (Line 437) What is the global restorativeness?
RESPONSE: We thank you for giving us the opportunity to specify this important part of our research. We previously defined “global restorativeness” the factor that integrates all the restorativeness subfactors: Being away, Fascination, Compatibility and Legibility, but we have now renamed it “general restorativeness” for consistency. As specified in section 3.2, in our research, the subfactor Coherence was instead considered as a separate aspect of restorativeness.
Point 28: (Line 442) Should it be Figure 2?
RESPONSE: Yes, we apologize for the mistake. We have now renamed “Figure 3” with “Figure 2”.
Point 29: (Line 443-451) This part is a duplication.
RESPONSE: We have now eliminated the duplication. Sorry for the mistake.
Point 30: (Figure 3) The title of the figure should be below the figure. Also, according to the information provided by the authors, two separate SEMs should be shown here.
RESPONSE: The title has been replaced as suggested. Concerning the suggestion of reporting separate SEMs, we preferred to keep both sets of analyses (i.e., on general Restorativeness and on Coherence) in the same model, and thus shown in the same figure. In fact, we believe that a crucial advantage of using SEM is the possibility of examining multiple dependent/endogenous variables within the same analytic framework, without the need to fit separate models.
Point 31: (Line 458) The title should be "Social and perceived environmental factors".
RESPONSE: Please note that we examined social factors along with objective ones within the same model, as shown in the subsection “Objective, social, and environmental factors”, and divided from perceived ones, named “Perceived environmental factors” (Subsection 3.4 Structural equation modelling).
Point 32: (Figure 4) Same comments as for Figure 3.
RESPONSE: Yes, we apologize for the mistake. We have now renamed “Figure 4” as “Figure 3”.
Point 33: (Line 505) Again, what is the global restorativeness?
RESPONSE: As you can read from the answer to point 27, we decided to replace “global restorativeness” with “general restorativeness” and keep it consistent throughout. As explanation of the meaning of “global restorativeness”, we indicated the factor that integrates the other subfactors of ART which are Being away, Fascination, Compatibility and Legibility but not Coherence. As specified in section 3.2, in our research, the subfactor Coherence was instead considered as a separate aspect of restorativeness. Therefore, we decided to add the adjective “global” or “general” to “restorativeness” to clarify that it integrates 4 subfactors, excluding Coherence.
Reviewer 2 Report
The article focuses on the relationship between some (perceived and objective) environmental/social conditions and perceived restorativeness, adapting for the first time an existing approach to a Mediterranean environment and offering a significant and original contribution to the sparse literature on the “bluespace effects”. The literature review on restorativeness is consistent and adequate, as well as a sound methodology and a solid final model are presented.
Nevertheless, I would invite the authors to take in account the following suggestions to further improve the article:
1) Given the special issue subject, the absence in the text of any reference to the SDGs, some of which are strongly linked to psychophysical well-being and health, is significant. I would like to invite the authors to further integrate these topics into the article, especially in the parts concerning the introductory premises and the conclusions (why are the results significant in the SDGs perspective?)
2) Perceived and objective environmental conditions (i.e., temperature) are the main focus of the research but the literature in this regard is only tangentially analyzed by the authors, while the topic of psycho-physical comfort associated with temperature and other environmental conditions has been largely addressed in the non-psychological literature. I would like to invite the authors to further expand their literature review in this direction. It could also provide useful references for their conclusions.
3) Lines 505-509: this correlation could be linked to a higher place attachment, as you suggested about crowding. Do the authors have an interpretation for this result with reference to the literature?
4) Lines 510-515: I would suggest looking at these results also considering the environmental self-regulation hypothesis (Kalevi Korpela), it may be useful for a more in-depth interpretation.
Author Response
- REVIEWER 2
The article focuses on the relationship between some (perceived and objective) environmental/social conditions and perceived restorativeness, adapting for the first time an existing approach to a Mediterranean environment and offering a significant and original contribution to the sparse literature on the “bluespace effects”. The literature review on restorativeness is consistent and adequate, as well as a sound methodology and a solid final model are presented. Nevertheless, I would invite the authors to take in account the following suggestions to further improve the article:
RESPONSE: We are grateful for this comment and for the important suggestions.
1) Given the special issue subject, the absence in the text of any reference to the SDGs, some of which are strongly linked to psychophysical well-being and health, is significant. I would like to invite the authors to further integrate these topics into the article, especially in the parts concerning the introductory premises and the conclusions (why are the results significant in the SDGs perspective?)
RESPONSE: We thank the reviewer for this comment. As suggested by the reviewer, we have now added some “key phrases” that premises to relate our study and our aims with SDGs perspective. We also added an integration of this important topic in our conclusions explaining why our results could help the expansion of knowledge about the 14th Sustainable Development Goal of 2030 agenda. Please see as follows:
“It’s clear that the pressure of modern-day westernized living is taking a toll on human’ quality of life and well-being (Vella‐Brodrick & Gilowska, 2022). Starting from this awareness, to mitigate these negative environmental effects on humans, actions focused on the changing the quality of life are needed (e.g., SDGS, 2015). This important topic has also inspired much environmental psychology research over the past 20 years, focusing on the role that the environment has on physiological, psychological, and emotional dimension of the individual” (Section 1. Introduction)
“According to the 14th Sustainable Development Goals of 2030 Agenda (SDG’s, 2015) to allow seas and oceans to contribute to the human well-being, it is important to preserve the physical and biological integrity of these particular and strongly vulnerable biomes” (Subsection 1.1 Blue space as a restorative environment)
“Furthermore, mass tourism and the lack of lack of control of human activities near the coasts are increasingly threatening marine habitat and their ability to supply important resources for humans (SDG’s, 2015). It is also demonstrated that environmental variables such as excessive noise, litter, or garbage can contribute to elicit psychological stress and can influence preference and perceived restorative quality in urban and natural environments (e.g., Wyles et al., 2016). Based on these premises, it is clear how changes in environmental quality and climate can affect perceived restorativeness in blue space. Increasing knowledge about this relationship could raise awareness that human well-being cannot be achieved without the protection of the earth's ecosystem (SDG’s, 2015)” (Subsection 1.1 Blue space as a restorative environment)
“As mentioned in the introduction, these current findings reinforce the validity of one of the most urgent topics in the SDGs perspective about marine environments and its vulnerability. It’s clear that the link between blue space and human well-being is not one-sided and it is often complex and most of the time human well-being is frequently generated at the cost biome’s integrity. For this reason, a change in how humans view, manage and use marine resources is strongly required (SDGs, 2015). As a conclusion to this work, we recommend to further implement research about blue spaces’ vulnerability to climate and environmental changes with methodological improvements and we hope that this scientific evidence can spread awareness about the importance of increasing local and regional sustainable projects, with the aim to promote blue space as a mental health and well-being resource for residents, tourists, and vulnerable people” (Section 5. Conclusions).
2) Perceived and objective environmental conditions (i.e., temperature) are the main focus of the research but the literature in this regard is only tangentially analyzed by the authors, while the topic of psycho-physical comfort associated with temperature and other environmental conditions has been largely addressed in the non-psychological literature. I would like to invite the authors to further expand their literature review in this direction. It could also provide useful references for their conclusions.
RESPONSE: Thank you for the suggestion. As requested, we have integrated this important topic in our manuscript. Please see as follow:
“Indeed, it is demonstrated that the increase of air temperature is negatively associated to individuals’ cognitive performance, through an important decline in both speed and accuracy cognitive measure (Yeganeh et al., 2018). From a physiological standpoint, temperature and air pollution seems to act together to alter the cardiac autonomic functions (Wu et al., 2013). Similarly, Ran et al., (2011) found that temperature may interact synergistically to affect heart rate variability (HRV). These findings demonstrate how climate change and environmental parameters may affect on both psychological and physiological dimensions of humans”.
“Past research found that the increase in air temperature causes physiological alterations on individuals (increase in blood pressure, low oxygen saturation in the blood). It is inferred that the observed physiological effects were mainly responsible for the negative effects on cognitive performance in the same sample (Li et al., 2020). It is also demonstrated that the increases in temperature have a negative effect on the emotional quality of an individual’s visit (e.g., Park et al., 2011). (Subsection 1.1 Blue space as a restorative environment)
“Importantly, we also found that perceiving better air quality increases the perceived restorativeness of visitors, which is in part consistent with Hipp & Ogunseitans’ findings. For further research, it would be interesting to cross psychological and physical outcomes of the increase of air temperature and air quality in blue space” (Section 5. Conclusions)
3) Lines 505-509: this correlation could be linked to a higher place attachment, as you suggested about crowding. Do the authors have an interpretation for this result with reference to the literature?
RESPONSE: Thank you for this suggestion. We agree that this topic is very important in our research. However, the variable "the number of people in the group that participants came with" is not related with the measurement of perceived crowding. Rather, it indicates whether participants came to the beach with at least another person, and in case with how many of them. For this reason, we decided to clarify that this result must be complemented by other research an in-depth to interpretation of the role of "other people in the same group" in restorativeness.
Please read as follow:
“These results could be linked to a higher perceived place attachment to people who visit the site more often. Considering previous literature, the sense of privacy and the need of control and security is important, especially in environments to which one is emotionally attached (Korpela, 1989). This has also been confirmed in more recent studies (e.g., Eder & Arnberger, 2012) that demonstrated that higher place attachment is associated to a minor tolerance to crowding. This would explain why perceived restorativeness decreases when the group of people that participant came with is bigger. Another possible explanation, strongly linked to the previous one, could be that other people in the group can be perceived as "distraction agents" for their own psychological restoration, but there is still no adequate research on this topic. Based on this, in future research it would be important to deepen understand the role of other people in the experience of perceived restorativeness”. (Section 4. Discussion)
4) Lines 510-515: I would suggest looking at these results also considering the environmental self-regulation hypothesis (Kalevi Korpela), it may be useful for a more in-depth interpretation.
RESPONSE: We thank the reviewer for these references that improve the discussion in our article. Beyond the premises that we have introduced in previous results, we added an interpretation taking into consideration to Self-regulation Hypothesis perspective as follows:
“According to Self-Regulation Theory, people seek environmental conditions, such as calm and secure places to regulate their affective and psychological states (Korpela, 1989). Visiting the beach in the winter months without crowds could be perceived as a more adequate environment for the self-regulation process" (Section 4. Discussion).
Round 2
Reviewer 1 Report
Dear authors,
Thanks for your responses to my comments and for revising them accordingly.
I think the current version can be accepted. However, please double-check some details of the manuscript before publication.
For example:
1) Table 1, which should be "NUMBER OF PEOPLE CAME WITH";
2) Line 233, 1= less thank 1 hour, 2= 1-2 hrs, 3= 2-4 hrs, 4= more than 4 hours;
3) Line 238, the author mentioned that "1 = alone, 2 = with another person, 3 = more than another person", yet in the table 1, the range of variable 6 is [1, 4].
4) Line 242, Table 1 does not contain any information about the activity. Or I missing something?
5) Line 353, although this is trivial detail, I guess R is not a software, but rather programming code?
6) Table 2, I appreciate the explanation and revision, however, I would like to suggest the authors to add the full name of TOT in the label of the table.
7) Line 433, from here on, the subsections are labeled incorrectly.
8) Line 469, I guess it should be Figure 2? The same as the title of the figure at line 473.
9) Please double-check all the numbering of the figures and tables. Also, the text about the numbering in the manuscript.
Congratulations and good luck.
Author Response
REVIEWER 1:
Dear authors,
Thanks for your responses to my comments and for revising them accordingly.
I think the current version can be accepted. However, please double-check some details of the manuscript before publication.
RESPONSE: we are grateful for the positive consideration of our paper. Note: all changes are marked in blue in the revised manuscript.
For example:
1) Table 1, which should be "NUMBER OF PEOPLE CAME WITH";
RESPONSE: Thank you for this note. As specified in subsection 2.3 Materials, “Number of people came with” in Table 1 refers to the question on “the number of people in the group that they came with” (With how many people have you come to the beach today? 1 = alone, 2 = with another person, 3 = more than another person). To make it clearer to the readers, we decided to rename the variable as “N OF PEOPLE IN THE GROUP” in table 1 and 3.
2) Line 233, 1= less thank 1 hour, 2= 1-2 hrs, 3= 2-4 hrs, 4= more than 4 hours;
RESPONSE: We thank the reviewer for this comment e we apologized for the mistake. We have now modified the text as recommended (subsection 2.3 Materials).
3) Line 238, the author mentioned that "1 = alone, 2 = with another person, 3 = more than another person", yet in the table 1, the range of variable 6 is [1, 4].
RESPONSE: We are sorry for the mistake. We checked our dataset: we confirm that it was a typing error on the manuscript, and the correct range is [1,3]. As recommended, we modified the range in Table 1.
4) Line 242, Table 1 does not contain any information about the activity. Or I missing something?
RESPONSE: That is correct. We have reported the frequencies in the Results, subsection 3.1: Characteristics of participants and visit. We apologize for not reporting any data in Table 1. We have now added some information about the activity in Table 1 (7. Relax Acivity).
5) Line 353, although this is trivial detail, I guess R is not a software, but rather programming code?
RESPONSE: We thank the reviewer for this specification. We modified as suggested (subsection 2.5 Data Analysis).
6) Table 2, I appreciate the explanation and revision, however, I would like to suggest the authors to add the full name of TOT in the label of the table.
RESPONSE: Thank you for this suggestion. it is not clear to us what “full name” of the subfactors is required. Anyway, to make clearer what we are referring to every subfactor total, we have reported the items of each single sub-factor of the PRS. Please see Table 2 and the note below.
7) Line 433, from here on, the subsections are labeled incorrectly.
RESPONSE: Thank you for noticing this. We have now modified the subsections correctly.
8) Line 469, I guess it should be Figure 2? The same as the title of the figure at line 473.
RESPONSE: Thank you for noticing the mistake in the numbering of the figures. We have now renamed the figures correctly (please see figure 2, figure 3 and figure 4).
9) Please double-check all the numbering of the figures and tables. Also, the text about the numbering in the manuscript.
RESPONSE: We thank the reviewer for this suggestion. We have now double checked the numbering of figures and tables and the related mentions in the text.